

# Single-cell transcriptome reveals the heterogeneity of malignant ductal cells and the prognostic value of REG4 and SPINK1 in primary pancreatic ductal adenocarcinoma

Yutian Ji[1,2,3], Qianhui Xu[4] and Weilin Wang[1,2,3]

[1] Department of Hepatobiliary and Pancreatic Surgery, The Second Affiliated Hospital, Zhejiang University School of Medicine, Hangzhou, Zhejiang, China
[2] Zhejiang University School of Medicine, Hangzhou, Zhejiang, China
[3] Key Laboratory of Precision Diagnosis and Treatment for Hepatobiliary and Pancreatic Tumor of Zhejiang Province, Hangzhou, China
[4] Fudan University, Shanghai, China

Corresponding author
Weilin Wang, wam@zju.edu.cn

## ABSTRACT

**Background.** Pancreatic ductal adenocarcinoma (PDAC) is one of the leading causes of cancer-related deaths, with very limited therapeutic options available. This study aims to comprehensively depict the heterogeneity and identify prognostic targets for PDAC with single-cell RNA sequencing (scRNA-seq) analysis.

**Methods.** ScRNA-seq analysis was performed on 16 primary PDAC and three adjacent lesions. A series of analytical methods were applied for analysis in cell clustering, gene profiling, lineage trajectory analysis and cell-to-cell interactions. *In vitro* experiments including colony formation, wound healing and sphere formation assay were performed to assess the role of makers.

**Results.** A total of 32,480 cells were clustered into six major populations, among which the ductal cell cluster expressing high copy number variants (CNVs) was defined as malignant cells. Malignant cells were further subtyped into five subgroups which exhibited specific features in immunologic and metabolic activities. Pseudotime trajectory analysis indicated that components of various oncogenic pathways were differentially expressed along tumor progression. Furthermore, intensive substantial crosstalk between ductal cells and stromal cells was identified. Finally, genes (REG4 and SPINK1) screened out of differentially expressed genes (DEGs) were upregulated in PDAC cell lines. Silencing either of them significantly impaired proliferation, invasion, migration and stemness of PDAC cells.

**Conclusions.** Our findings offer a valuable resource for deciphering the heterogeneity of malignant ductal cells in PDAC. REG4 and SPINK1 are expected to be promising targets for PDAC therapy.

# INTRODUCTION

Pancreatic ductal adenocarcinoma (PDAC), one of the most aggressive and fatal diseases with a current 5-year survival rate less than 10%, accounts for approximately 90% of malignant pancreatic diseases (*Siegel et al., 2022*). Surgical resection remains the main choice for PDAC patients. However, most of them are diagnosed at advanced stage, losing the opportunity to receive a surgery (*Winter et al., 2012*; *Gobbi et al., 2013*; *Ryan, Hong & Bardeesy, 2014*). Although multiple novel therapies have been beneficial with different types of solid tumors, very few treatment modalities have shown promising efficacy in PDAC (*Tempero et al., 2019*; *Sohal et al., 2021*; *Hosein et al., 2022*). A primary reason for the observed treatment recalcitrance and high mortality rate is owing to its complicated intratumoral heterogeneity and intensive cellular crosstalk (*Moffitt et al., 2015*; *Ho, Jaffee & Zheng, 2020*).

Previous genomic and transcriptomic studies have revealed that critical gene mutations or aberrant signaling pathways driving the pathogenesis of PDAC, such as KRAS (over 90%), TP53, SMAD4 and CDKN2A (over 50%) (*Australian Pancreatic Cancer Genome Initiative et al., 2012*; *Witkiewicz et al., 2015*). Other novel recurrent mutations (<10%) have also been identified from unbiased analyses of PDAC. These diverse mutations converge on specific pathways and processes, including KRAS, TGF-$\beta$, WNT, Notch signaling, chromatin remodeling and DNA repair pathways (*Australian Pancreatic Cancer Genome Initiative et al., 2015*; *Raphael et al., 2017*). Also, deep insights have been gained regarding the immunologic and metabolic reprogramming in PDAC progression (*Yao, Maitra & Ying, 2020*). The diversity of stromal and immune cell types comprises a quite complicated tumor microenvironment (TME) in PDAC (*Teng et al., 2015*; *Ligorio et al., 2019*; *Ho, Jaffee & Zheng, 2020*; *Steele et al., 2020*). Meanwhile, prominent metabolic adaptions, usually reflected as the 'Warburg effect', are one of the hallmarks of pancreatic cancer cells (*Warburg, Wind & Negelein, 1927*). Except for the enhanced glycolysis, multiple other metabolic activities, like fatty acid metabolism, steroid biosynthesis and glutamine synthesis, are largely altered in PDAC as well (*Daemen et al., 2015*; *Karasinska et al., 2020*).

In the past decades, some major breakthroughs in PDAC treatment have been made for a small fraction of patients based on bulk mRNA sequencing (*Oettle et al., 2013*; *Heining et al., 2018*; *Katz et al., 2021*). Unfortunately, most PDAC patients have not benefited from our current knowledge of the genetics and biology on PDAC yet, due to the limited picture of cellular complexity provided by bulk profiling technologies (*Yao, Maitra & Ying, 2020*). Recent advances in single-cell genomics have enabled us to conduct in-depth analyses of tumoral heterogeneity at unprecedented molecular resolutions (*Tanay & Regev, 2017*). It has effectively recognized multiple evolutionary lineages and cellular subpopulations, as well as their relative crosstalk within the TME. To date, a series of novel molecular subtype classifications has been developed for PDAC based on scRNA-seq (*Ligorio et al., 2019*; *Elyada et al., 2019*; *Hosein et al., 2019*; *Peng et al., 2019*; *Lin et al., 2020*; *Lee et al., 2021*), which has complemented the original binary classification including classical and basal-like subtypes (*Collisson et al., 2019*). In addition to tumor cells, other stromal cells

shaping the TME complexity were also classified into different components with specific gene expression patterns.

Here, we applied scRNA-seq analysis to dissect the intratumoral heterogeneity during PDAC progression. The transcriptome profile of a total of 32,480 cells from 16 primary PDAC tumors and three adjacent tissues was acquired. We found that PDAC tumor mass was highly heterogeneous and composed of diverse malignant and stromal cell types. Further, malignant ductal cells were distinguished into five subgroups by featured gene expression and biological profiles. Significant gene expression alterations which related to known tumor-related signaling pathways were identified as well. In addition, we also described intricate multicellular crosstalk in the TME. Therefore, our study delineates a comprehensive understanding of the single-cell transcriptome landscape in human primary PDAC and may hopefully provide a resource for further investigations aimed at characterizing and targeting specific populations in PDAC.

## MATERIALS & METHODS

### Information of single-cell datasets

The sequencing information of 32,480 single cells from 16 treatment-naive primary PDAC tumors and three adjacent tissues (detailed in Tables S1, S2) was acquired from the Gene Expression Omnibus (GEO) database (GSE155698) (*Steele et al., 2020*). Detailed parameters of gene-barcode matrices, feature data, as well as unique molecular identifier (UMI) count tables were illustrated in published works (*Li et al., 2017*). All information applied in our study is publicly available and open access.

### ScRNA sequencing data processing

Data processing of scRNA sequencing in our study was performed as described previously (*Xu et al., 2021a*). Briefly, Seurat (v2.3.0) R toolkit with the Read 10× () function was applied to import the Seurat object containing gene expression data (*Satija et al., 2015*). Quality control was conducted before the subsequent analysis of scRNA-seq data. All functions were run with default parameters, unless specified otherwise. Gene-cell matrices were filtered to exclude unqualified cells (<500 transcripts/cell, >30% mitochondrial genes) and genes (<300 cells/gene and >25,000 cells/gene). Each sample was represented as a fraction of the gene multiplied by 10,000, then transformed into a natural logarithm and normalized after adding one to avoid taking the logarithm of zero.

### Cell clustering and annotation

The Seurat package implemented in R was applied to identify major cell types. Principal component analysis (PCA) was conducted using the top 2,000 highly variable genes (HVGs) derived from the normalized expression matrix (Fig. S3A). Principal components (PCs) whose estimated $P$ value less than 0.05 were selected and the optimal resolution was determined by the cluster tree algorithm (Fig. S3B). Additional screening methods (DoubletFinder and CellCycleScoring) were performed to exclude unqualified cells for subsequent analysis (Figs. S4A, S4B). t-distributed Stochastic Neighbor Embedding (t-SNE) analysis was performed to visualize the clustering of single cells. The cluster distribution

of single cells from each sample was presented (Fig. S4D). Genes specific for each cluster were determined with FindAllMarkers based on the normalized gene expression data. 'Find.markers' was conducted in identifying the DEGs in different clusters. DEGs and literature-known markers were used to define the cell groups (Table S3).

## CNV assessment in single cells

InferCNV R package (version 1.4.0) was applied to estimate CNV of single cells (*Patel et al., 2014*). Various parameters were used to evaluate the inferCNV analysis, including "denoise", default hidden Markov models (HMMs), and a value of 0.1 for "cutoff". The default Bayesian latent mixture model with a threshold of 0.0005 was performed to assess the posterior probabilities of CNV alterations to minimize false-positive CNV calls.

## Identification of DEGs and pathways

Wilcoxon Rank-Sum Test with FindMarkers function (adjusted *P*-value <0.05, only.pos = TRUE and logFC.threshold = 0.25) was used to determine the DEGs within each cluster. Analysis for alterations in hallmark oncogenic and metabolic pathways was predominantly conducted with MSigDB databases (https://www.gsea-msigdb.org/gsea/msigdb) (*Liberzon et al., 2011*). Then, the gene set variation analysis (GSVA) package (version 1.36.3) (*Hänzelmann, Castelo & Guinney, 2013*) was used to estimate relative pathway activities in different ductal cell clusters.

## Pseudotime trajectory analysis for single cells

Using Monocle2 (v2.16.0) (*Trapnell et al., 2014*), single-cell pseudotime trajectory analysis was performed to investigate cell state transitions under the assumption that one-dimensional 'time' could delineate the high-dimensional expression values. The previously identified malignant cell cluster was loaded into R environment. The newCellDataSet function was conducted to create an object with the parameter expressionFamily = negbinomial.size. Genes qualifying for our standards (mean_expression $\geq$ 0.1 and dispersion_empirical $\geq$ 1 * dispersion_fit identified) were filtered for the trajectory analysis. Dimension reduction was achieved using reduceDimension() with parameters reduction_method = "DDRTree" and max_components = 2. Meanwhile, minimum spanning trees were plotted using visualization function "plot_cell_trajectory". "DifferentialGeneTest" and "plot_pseudotime_heatmap" were applied respectively to estimate and visualize gene expression changes following the pseudotime trajectory, and then genes were grouped into subclusters based on their expression profiles. Gene expression patterns (detailed in Table S4) were utilized to group genes into subgroups. Genes derived from Branch Expression Analysis Modeling (BEAM) analysis (with a *q*-value <0.05) were grouped and plotted with "plot_genes_branched_heatmap()" function (detailed in Table S5).

## Cell-to-cell interaction analysis

Cells defined as malignant ductal cells, macrophages, T cells, B cells, dendritic cells, monocytes, NK cells and cancer-associated fibroblasts (CAFs) in our analysis were input to CellPhoneDB (version 2.1.7) for exploring potential cell-to-cell interactions among

them (*Efremova et al., 2020*). Then, receptors and ligands respectively expressed in more than 10% of cells in the corresponding subclusters were identified as the most relevant cell type-specific interactions.

As for pairwise comparisons, cluster labels of all cells were firstly permuted randomly for 1,000 times to obtain the average expression levels of the receptors and ligands among interacting clusters, after which a null distribution was generated for each receptor–ligand pair. Then, ultimate results for cell-to-cell interactions were determined by measuring *P* values, which were obtained by comparing the proportion of the means that were higher than the actual mean, for the probability of the cell-type specificity of the corresponding receptor and ligand.

## Validation of gene expression

Three primary human pancreatic cancer cell lines (BXPC-3, CAPAN-2 and Mia-PACA2) and the normal pancreatic ductal cell line (HPNE) were purchased from the Cell Bank of the Type Culture Collection of the Chinese Academy of Sciences, Shanghai Institute of Biochemistry and Cell Biology. The purity of cells used in our experiment was verified by short-tandem repeat (STR) polymorphism analysis and mycoplasma was regularly tested to avoid mycoplasma contamination. BXPC-3 and CAPAN-2 were cultured in RPMI 1640 (Cat. No. 11875093; Gibco, Waltham, MA, USA), while MIA-PACA2 and HPNE were cultured in DMEM (Cat. No. 11965092; Gibco, Waltham, MA, USA), both with 10% fetal bovine serum (FBS; Cat. No. 16000-044; Gibco, Waltham, MA, USA). Cells were cultured in a humidified atmosphere of 5% $CO_2$ and 95% relative humidity at 37 °C.

Quantitative real-time PCR (qRT-PCR) was performed for mRNA quantification. qRT-PCR results in triplicates were analyzed as described previously (*Xu et al., 2021b*). The relative expression of potential prognostic genes was measured with the $2^{-\Delta\Delta Ct}$ method, with 18S levels as an endogenous control. Primer sequences of above genes were listed: REG4, 5′-TGAGGAACTGGTCTGATGCCGA-3′ and 5′-TCCATATCGGCTGGCTTCTCTG-3′; SPINK1, 5′-ATGACCCTGTCTGTGGGACTGA-3′ and 5′-GCGGTGACCTGATGGGATTTCA-3′; and 18S, 5′-ACCCGTTGAACCCCATTCGT GA-3′ and 5′-GCCTCACTAAACCATCCAATCGG-3′.

As for protein levels, cells were lysed in lysis buffer (Cat. No. G2002; Servicebio, Beijing, China) in the presence of protease inhibitor cocktail (Cat. No. P2714; Sigma, Burlington, MA, USA). Cell lysates were then subjected to western blotting as previously described (*Cui et al., 2020*). Primary antibodies were used as follows: REG4 (Cat. No. ab255820; Abcam, Cambridge, UK), SPINK1 (Cat. No. ab183034; Abcam, Cambridge, UK), and GAPDH (Cat. No. 10494-1-AP; Proteintech, Chicago, IL, USA). SuperSignal West Pico PLUS Chemiluminescent Substrate (Thermo Fisher, Waltham, MA, USA, Cat. No. 34577) was used for chemiluminescence staining. Bio-Rad ChemiDoc Imaging System was utilized for chemiluminescence detection.

In addition, Kaplan–Meier survival analysis was detected with Gene Expression Profiling Interactive Analysis (GEPIA, http://gepia.cancer-pku.cn) (*Tang et al., 2017*). Immunohistochemistry (IHC) images were obtained from Human Protein Atlas (HPA, https://www.proteinatlas.org/) (*Uhlén et al., 2015*).

## Analysis for immune infiltration

Tumor Immunity Estimation Resource (TIMER, https://cistrome.shinyapps.io/timer) (*Li et al., 2016*) was used to analyze the correlation between gene expression and immune infiltrating cell types. The purity-corrected partial Spearman's correlation coefficient was used to evaluate the relationship between gene expression and immune infiltration.

## Cell transfection

pGMLV-REG4-shRNA-puro and pGMLV-SPINK1-shRNA-puro plasmids were purchased from Genomeditech (Shanghai, China). HEK-293T cells were transfected with psPAX2, pVSVG and lentiviral plasmids to produce lentiviruses. Virus particles were collected from supernatants and filtered through a 0.45 μm filter. MIA-PACA2 cells were infected by indicated viruses, followed by puromycin selection for 3 weeks, to construct stable cell lines.

## Cell proliferation assay

Cell Counting Kit-8 (CCK-8, MCE, Cat. No. HY-K0301) was used to detect cell proliferation. Cells after transfection were seeded in 96-well plates at a density of 4000 cells/well, and incubated overnight for adherence. Cells were cultured in the presence/absence of 50 nM gemcitabine hydrochloride (MCE, Cat. No. HY-B0003). CCK-8 reagent was added into wells after 0, 24, 48, 72 h, following incubation for 2 h at 37 °C, as recommended by instructions. Cell absorbance was detected with a microplate reader at 450 nm.

## Colony formation assay

Cells were seeded in 6-well plates at a density of 1,000 cells/well. Then they were cultured for 10 days in the incubator, with culture medium replacement every 3 days. At day 10, cell colonies were fixed with 4% paraformaldehyde for 10 min and stained with 0.1% crystal violet for 15 min. The number of colonies (>50 cells per colony) was counted under microscopy.

## Wound healing assay

To examine cell migration, cells were seeded in 6-well plates at a density of $6 \times 10^5$ cells/well. Then they were cultured in the incubator until cells were covered almost confluent. The surface of cells was scratched by the tip of a 200 μL pipette and washed twice with PBS. Cell images were captured with microscopy immediately and 20 h later.

## Sphere formation assay

Tumor spheroids were created with a hanging drop method (*Honeder et al., 2021*). Cells after transfection were suspended at a concentration of $5 \times 10^4$ cells/mL in culture medium containing 3 μg/mg type I collagen (Cat. No. 354236; Corning, Corning, NY, USA). 20 μL droplets were transferred to the lid of 100 mm dishes and were inverted over dishes containing 5 mL phosphate buffer solution (PBS) to avoid drying. About 8 days later, spheroids were observed under microscopy and their diameters were measured.

### Data analysis

All statistical analyses for scRNA-seq data were carried out with R (http://www.r-project.org). Not each of the data points was presented in all box and violin plots due to the fact that extensive data points would obscure overall distribution. A two sided paired or unpaired Student's *t*-test and unpaired Wilcoxon rank-sum test were performed in the cases indicated. For *in vitro* experiments, data from three independent experiments were expressed as the mean $\pm$ SEM and analyzed using GraphPad Prism 7. The relative protein levels were measured with Fiji Is Just Image J (FIJI, win32). Statistical significance was defined as a *P* value <0.05.

## RESULTS

### Single-cell atlas in primary PDAC

To understand the diversity in the TME of PDAC, 16 samples diagnosed with primary PDAC and three adjacent pancreatic tissues were included for scRNA-seq analysis. For unbiased clustering, 26 main cell clusters were identified at first (Table S3). The cluster distribution of single cells from each patient was presented (Figs. S3C, S3D). According to cell expression patterns and known markers, cells were further optimized to six clusters with differential gene expression profiles, including ductal cells, myeloid cells, T cells, B cells, NK cells and CAFs (Fig. 1A). In most cases, multiple well-known markers were used to identify cell clusters, such as KRT19 and TSPAN8 for ductal cells, AIF1 and FPR1 for myeloid cells, CD2 and CD3 for T cells, CD79 for B cells, NKG7, GNLY, KLRD1 for NK cells, and COL1A1 and COL1A2 for CAFs (Fig. 1B). In general, we observed a distinct increase in the proportion of ductal and B cell clusters in PDAC samples compared to adjacent tissues, while a decrease in the proportion of NK cells (Fig. 1C). Meanwhile, 16 PDAC samples shared similarities in cell type composition, while the relative proportion of each cell cluster exhibited a big discrepancy (Fig. 1D), which reflects both the tumoral heterogeneity and similarity across PDAC patients.

### CNV landscape distinguishing malignant ductal cells in PDACs

To investigate malignant status of cells, large-scale chromosomal CNVs were defined in single cells by inferCNV. A clustered heatmap was generated across samples (Fig. 2A). Ductal cells exhibited remarkably higher CNV levels than other cell types. CAFs presented moderate CNV levels. CNVs in ductal cells and CAFs were both significantly higher in PDAC than those in adjacent tissues (Figs. 2B, 2C). These results demonstrated that ductal cells were the most predominantly malignant cells in PDAC, which were also supported by the specific expression of poor prognosis markers KRT19 and TSPAN8 mentioned above. Ductal cells in PDAC samples expressing relatively high CNVs, with immune cells as references, were screened out and defined as 'malignant cells' for subsequent analysis.

### Distinct subgroups in malignant cells

Malignant cells identified above were further analyzed and clustered into five subtypes according to t-SNE analysis (Fig. 3A, Fig. S5A). By comparing gene expression levels, we found that each subgroup expressed its specific gene set which could be distinguished from
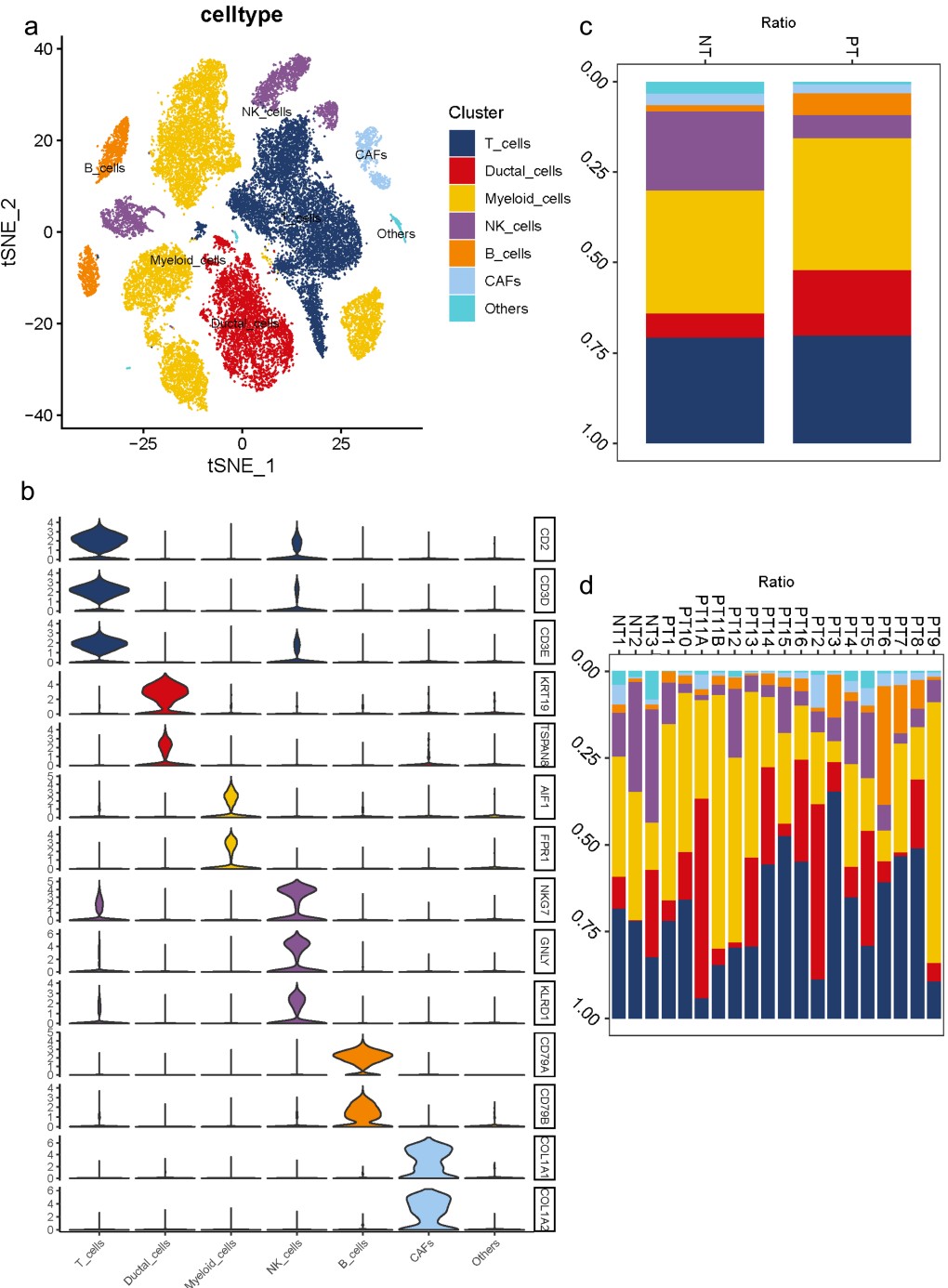

**Figure 1  Diverse cellular subclusters in primary PDAC analyzed by single-cell transcriptomics.** (A) Six main cell types identified in primary PDAC lesions. (B) Signature genes across six subtypes presented on violin plots. (C) Cell cluster proportions between adjacent normal tissues and primary PDAC lesions. (D) Cell cluster proportions across 19 samples.

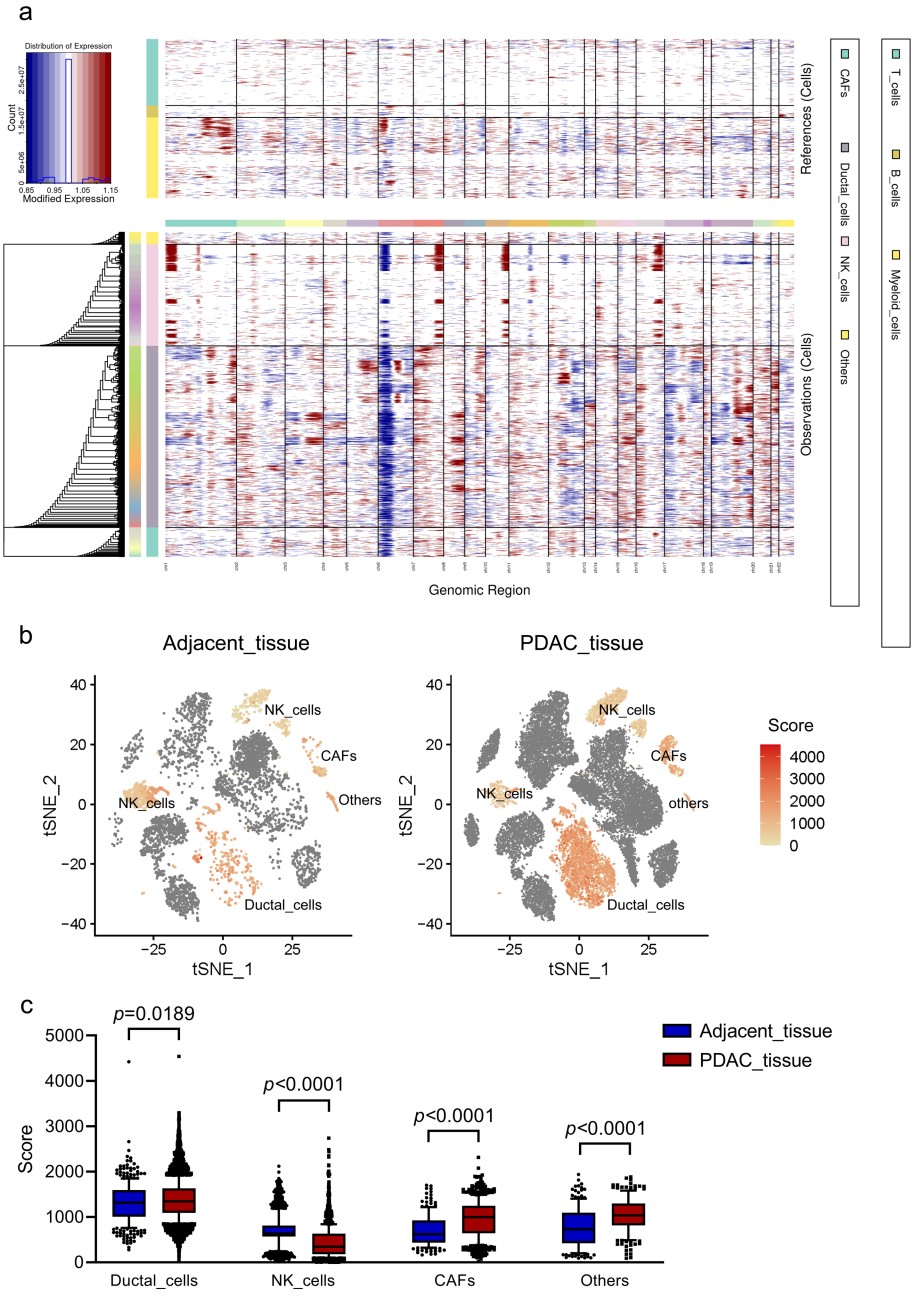

**Figure 2  CNV analysis in different cell clusters in primary PDAC.** (A) A hierarchical heatmap of large-scale CNVs of 16 PDAC samples and 3 adjacent normal tissues. (B) Specific distributions of CNV scores in different cell clusters. (C) CNV scores in cell clusters depicted by box plots.

each other. Notably, subgroup 1 was the major population presented in malignant cells with high levels of MMP7 and DEFB1; subgroup 2 exhibited high levels in CEACAM5 and CLDN18; subgroup 3 were mainly characterized by CA2 and GPX2; subgroup 4 specifically expressed FXYD2 and AMBP; and RGS13 and AZGP1 were uniquely expressed

in subgroup 5 (Fig. 3B). The specific distribution of each cell marker expression in single cells was presented (Fig. 3C).

To further describe the biological functions of malignant subgroups, GSVA was conducted in the following analysis (Fig. 3D, Fig. S5B). Generally, genes from multiple classical oncogenic pathways were enriched in almost each subgroup (Hezel et al., 2006), further confirming their malignancy. In particular, subgroup 1 was enriched for genes from ERBB signaling pathway. Genes from MYC, WNT/$\beta$-catenin and PI3K/AKT/MTOR pathways were enriched in subgroup 2. Subgroup 3 was enriched for MAPK and Notch pathways. Subgroup 5 was enriched for genes from P53 signaling pathway. In addition, gene expression patterns in subgroup 2, 3 and 4 were also related to KRAS pathway, which is the most frequently mutated pathway in PDAC. Also, commonly reported cellular biological and metabolic activity alterations were also detected in each subgroup specifically (Yao, Maitra & Ying, 2020). For instance, subgroup 1 showed heightened activities of unfolded protein response (UPR) and reactive oxygen species (ROS) pathway. Genes for subgroup 2 exhibited high enrichment involved in glycose and lipid metabolism, such as glycolysis and adipogenesis, which suggested the elevated energy requirements during progression. Additionally, cellular antioxidant activities like selenoamino acid metabolism and sulfur metabolism were also enriched in subgroup 2. Genes in subgroup 3 were enriched for autophagy regulation and epithelial mesenchymal transition (EMT). Upregulated genes in subgroup 5 were involved in angiogenesis.

Notably, we observed that subgroup 2 and 3 also involved in immune regulation. Specifically, subgroup 2 mainly participated in B cell and T cell signaling pathways, as well as interferon response, while subgroup 3 mostly involved in innate immune response including NOD-like and REG-I-like receptor signaling pathways.

Recently, accumulating studies have revealed essential roles of diabetes and insulin resistance in PDAC initiation and progression (Zhang et al., 2019; Deng et al., 2022). Interestingly, we also observed that gene functions in subgroup 1 and 2 were related to diabetes mellitus and insulin signaling pathway, which indicated their potential regulations in PDAC progression through mediating diabetes and insulin signaling pathways.

Taken together, these findings indicated that ductal cells in PDAC lesions consistently exhibited tumor malignancy patterns, while each subgroup executed its specific and diverse functions to fit together to mediate PDAC progression.

## Gene expression patterns of malignant cells during PDAC progression

To further determine which subgroup in malignant cells is responsible for tumor initiation, pseudotime trajectory analysis was performed across them. Using R package monocle, the lineage differentiation trajectory was constructed as a tree-like structure (Fig. 4A, Fig. S6). Cells encountered four nodes and eight states during the trajectory progression: pre-branch (State 1, 2, 3, 8, 9) and the other two main progressive branches named branch 1 (State 4, 5, 6) and branch 2 (State 7) (Fig. 4B). Subgroup 4 and 5 mainly appeared at the beginning of the trajectory. Subgroup 2 predominantly appeared in the progressive state. Notably, there were several subgroups showing their abilities in transition. For instance, subgroup

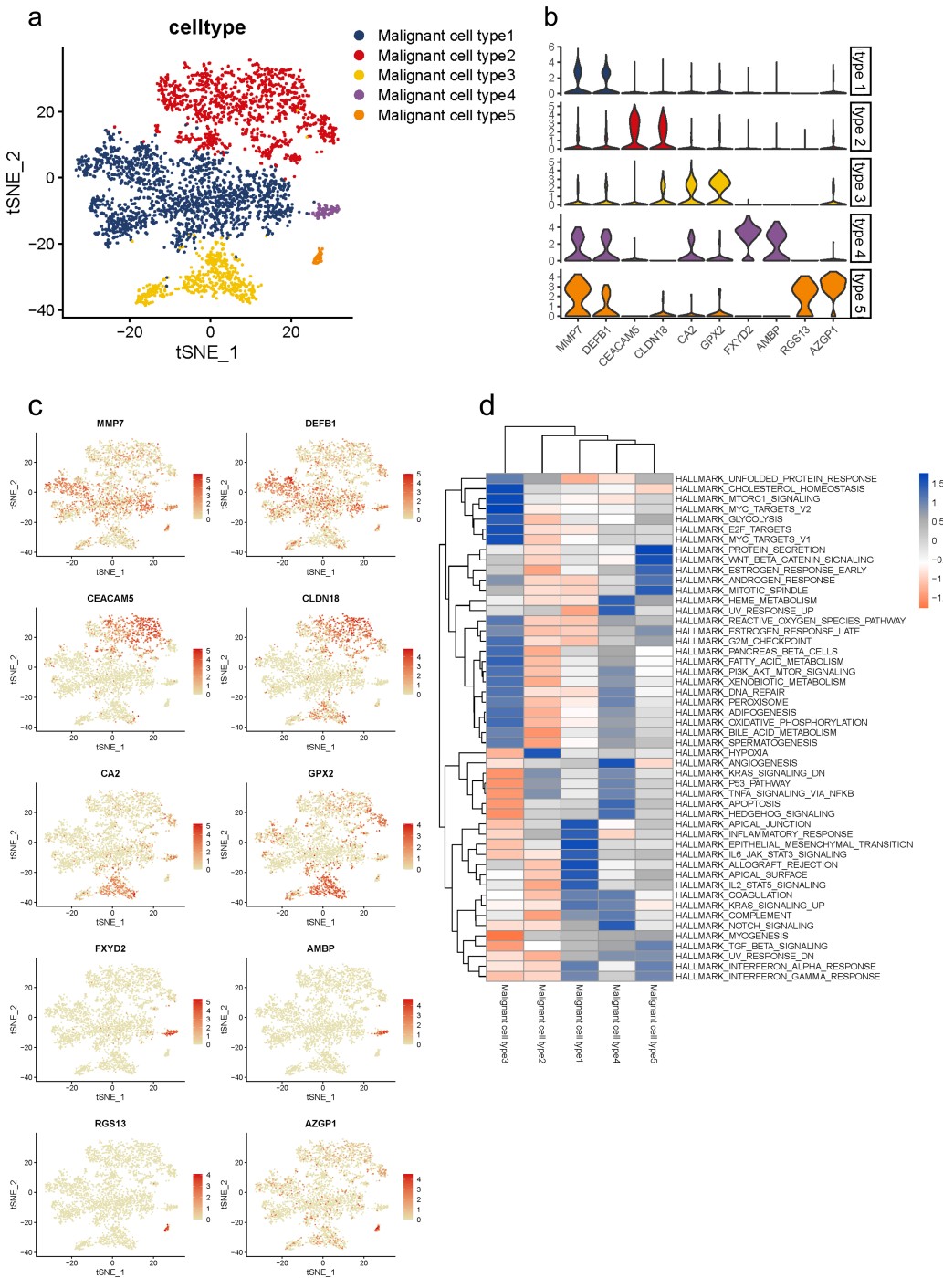

**Figure 3  Distinct subclusters of malignant ductal cells in primary PDAC lesions.** (A) a t-SNE plot showing five malignant ductal subclusters from 16 PDAC samples. (B) Violin plots for marker genes of each subcluster. (C) Feature plots for marker genes of each cluster. (D) Enrichment of Hallmark pathway terms in each subgroup.

1 and 3 mainly distributed throughout the beginning and branch 2, which showed a specific transitional direction from the beginning to branch 2. Subgroup 2 predominantly presented a specific transitional direction from the beginning to branch 1. Although these subgroups showed a preference in transition, they were detected at the terminal points of both branches, indicating their diverse forms along with the progression (Figs. 4C, 4D).

Along with the trajectory, cancer stem cell (CSC) marker ALDH1A1 persistently maintained a high level during transitions (*Lee, Dosch & Simeone, 2008*; *Sergeant et al., 2009*). During the transition to branch 1, cells sustained high expression levels of multiple reported poor prognosis PDAC markers like CEACAM5, HSPA6 and KLF4 (Fig. 4E) (*Deane & Brown, 2018*; *Gazzah et al., 2022*; *He, He & Xie, 2023*), which decreased during transition to branch 2. In addition, REG4, which is correlated with advanced stage, as well as one of the insulin-like growth factors, IGFBP3, both showed an upward trend in the transition to both branches (*Hwang et al., 2020*).

In addition, 5,633 genes whose expression significantly altered during transitions were analyzed (Table S4). In particular, a large portion of genes presenting malignant expression patterns was predominantly dysregulated along with the trajectory (Fig. 4F), including HMGA1, HMGB2, HES1, HIF1A, KLF2/4, ID1, FOS and P53, which were related to cell proliferation, EMT, DNA repair and hypoxia stress. Meanwhile, the alteration of multiple crucial oncogenic pathways (*Tempero et al., 2019*; *Yao, Maitra & Ying, 2020*), including Notch and PTEN signaling pathways, were remarkably activated. With featured gene expression patterns, we also found prominent changes in genes like CD55, HSP90AB1, REG4 and SPINK1 during the trajectory (Figs. 5A, 5B). Following the branched heatmap with BEAM analysis, genes were allocated to different clusters according to their expression features, and with the most divergent genes identified. These genes included S100A10, CEACAM5, NEAT1, MALAT1, SPINK1, REG4 and VEGFA (Fig. 5C).

## Intercellular interactions in the TME

To better present a comprehensive and diverse landscape of interactions between malignant ductal cells and various stromal cells, we performed further subclustering of T cells and myeloid cells with known markers, since they are the major populations shaping immune landscape within the TME (*Ho, Jaffee & Zheng, 2020*). Specifically, T cells were subdivided into CD4[+] T cells (marked by CD4, SELL), CD8[+] T cytotoxic cells (marked by CD8, GZMB) and CD8[+] T exhausted cells (marked by CD8, TIGHT; Figs. S7A–S7C). Myeloid cells were subdivided into monocytes (marked by IL1B, FPR1), dendritic cells (marked by CLU, AREG) and macrophages (marked by CD163, CD86; Fig. S7D–S7F).

An analysis of the interactions between different cell types in the TME was generated by CellPhoneDB. Results of the intercellular ligand–receptor interactions could be visualized and analyzed through heatmap plots. It suggested that macrophages, CD8[+] T cells, monocytes and NK cells, had the largest number of interacting pairs and improved pairwise communications, consisting the core nodes of cell-to-cell interactions (Fig. S8). It was also determined that various cell types interacted extensively, which was based on broadcast ligand–receptor pairs we detected. (Figs. 6A, 6B).

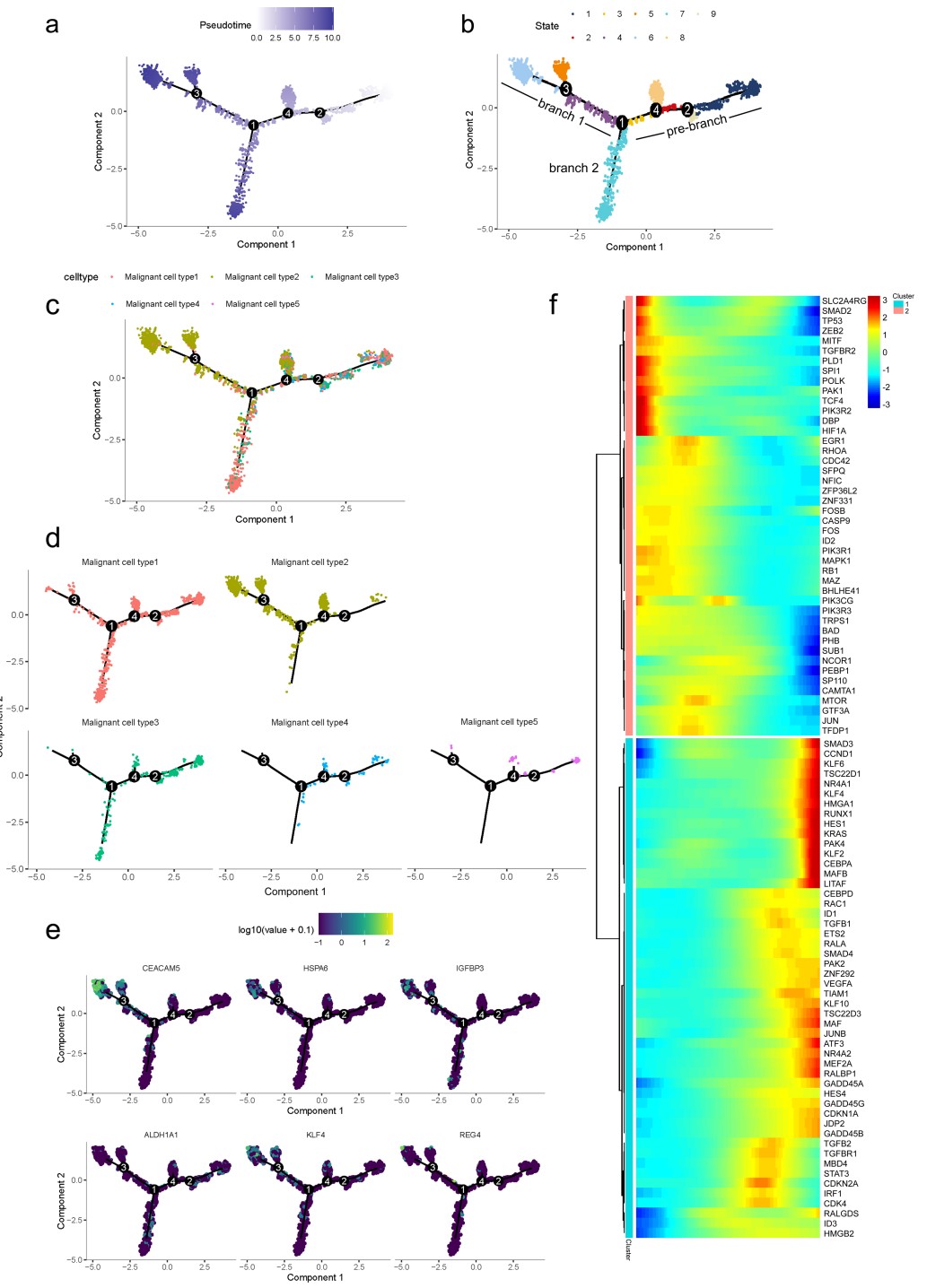

**Figure 4 Differential gene expression profiles along malignant progression.** (A) Pseudotime trajectory analysis of malignant ductal cells. (B) Cell states and branches, (C) distribution of five malignant subtypes and (D) distribution of each malignant subtype in pseudotime trajectory. (E) A single-cell trajectory plot showing expression of representative genes. (F) Expression of representative PDAC-associated genes across single cells depicted in a heatmap.

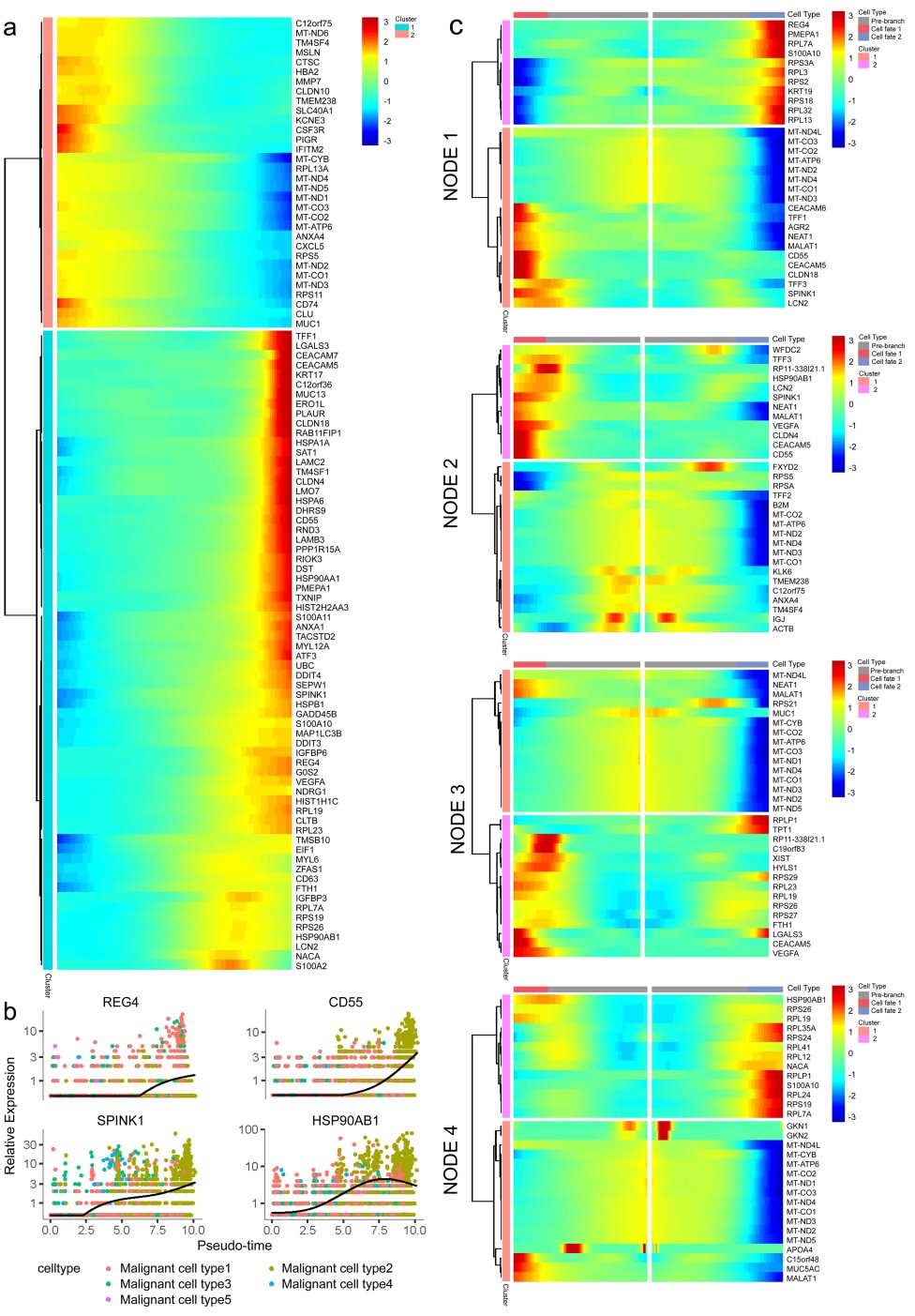

**Figure 5** **Profiling of gene expression during progression of malignancies.** (A) A hierarchical heatmap clustering of DEGs following the pseudotime curve. (B) Representative expression patterns of genes during progression. (C) A pseudotime heatmap displaying DEGs between different branches based on the BEAM analysis.

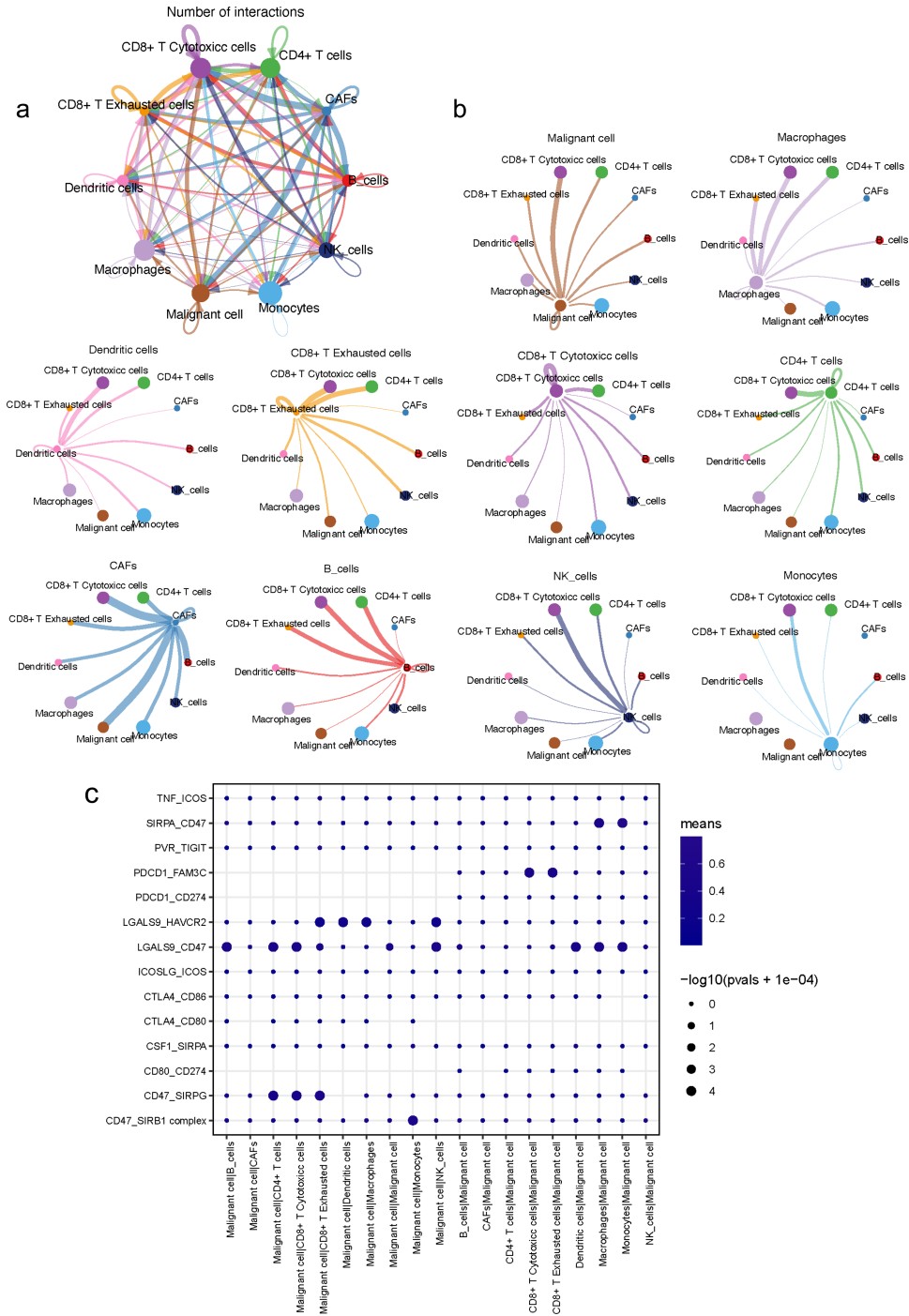

**Figure 6 Intensive crosstalk between malignant ductal cells and stromal cells in the TME.** (A) Intercellular crosstalk between malignant cells and stromal cells. (B) Detailed information regarding the ligands expressed by each subtype, as well as cells expressing cognate receptors. (C) A summary of selected ligand–receptor interactions in malignant cells under inhibitory interactions.

Meanwhile, an investigation of the inhibitory ligand–receptor complex between malignant ductal cells and other cell types was conducted. In addition to the well-defined inhibitory receptor–ligand pairs SIRPA-CD47, PVR-TIGHT and LGALS9-HAVCR2 (*Kučan Brlić et al., 2019*; *de Mingo Pulido et al., 2021*; *Jiang et al., 2021*), we found novel interactions between malignant cells and surrounding immune cells which were poorly investigated before, like PDCD1-FAM3C, LGALS9-CD47 and CSF1-SIRPA (Fig. 6C).

A wide range of costimulatory interactions, including known complexes like TNFRSF1A-GRN, MIF-TNFRSF14 and LGALS9-CD44, have also been discovered between malignant cells and other cell types (Fig. 7A). Interestingly, intensive interactions were found between macrophages and malignant cells, with four of the most commonly reported ligand–receptor pairs CD74-MIF, SPP1-CD44, CD74-COPA and CD74-APP, whose dysregulations are associated with tumor initiation and metastasis (*Stein et al., 2007*; *Orian-Rousseau, 2015*). These results indicated a pivotal role of macrophages through intensive functional interactions with other cells in PDAC progression, which is consistent with previous studies (*Vitale et al., 2019*). In addition, malignant cells also had common interactions with B cells, like CD74-COPA, CD74-MIF and CD74-APP. Meanwhile, we detected broad chemokine and receptor interactions between malignant cells and other cells (Fig. 7B), including CCR6-CCL20, CXCR3-CCL20 and CXCR6-CXCL16 (*Romero et al., 2020*), suggesting the chemoattraction potential of malignant ductal cells in the formation of a tumor-promoting microenvironment.

## Validation of potential prognostic targets *in vitro*

Two potential prognostic genes, regenerating family member 4 (REG4) and serine peptidase inhibitor Kazal type 1 (SPINK1), were selected from DEG results aforementioned (Figs. 4 and 5). REG4 is known as a member of the calcium-dependent lectin gene superfamily, which aberrantly expresses in multiple cancers including PDAC (*Hwang et al., 2020*; *Bishnupuri et al., 2022*). SPINK1 is a protease inhibitor of trypsin in the pancreas, whose mutation usually relates to chronic pancreatitis (*Chen et al., 2018*; *Suzuki & Shimizu, 2019*; *Suwa et al., 2021*). These genes were reported to be closely related to PDAC malignant phenotypes by multiple studies, while mechanisms of which are still need to be further elucidated.

To investigate mRNA and protein levels of them, qRT-PCR and western blotting were performed on commonly used primary PDAC cell lines, BXPC-3, CAPAN-2 and MIA-PACA2, as well as a normal pancreatic ductal cell line HPNE as a control. We generally observed significant upregulated expressions in REG4 and SPINK1 at both mRNA and protein levels in all PDAC cell lines compared to HPNE (Figs. 8A, 8B). In addition, differentially expressed protein levels of them were observed by IHC staining of normal pancreas and PDAC tissues obtained from HPA database (Fig. S9). These results were consistent with our findings in scRNA-seq mentioned above.

Another important finding was the correlation between gene expression and immune infiltration in PDAC. TIMER analysis revealed that SPINK1 significantly correlated with a wide range of immune cell types (including CD4[+] T cells, CD8[+] T cells, macrophages, neutrophils and dendritic cells) and multiple immune inhibiting molecules (including

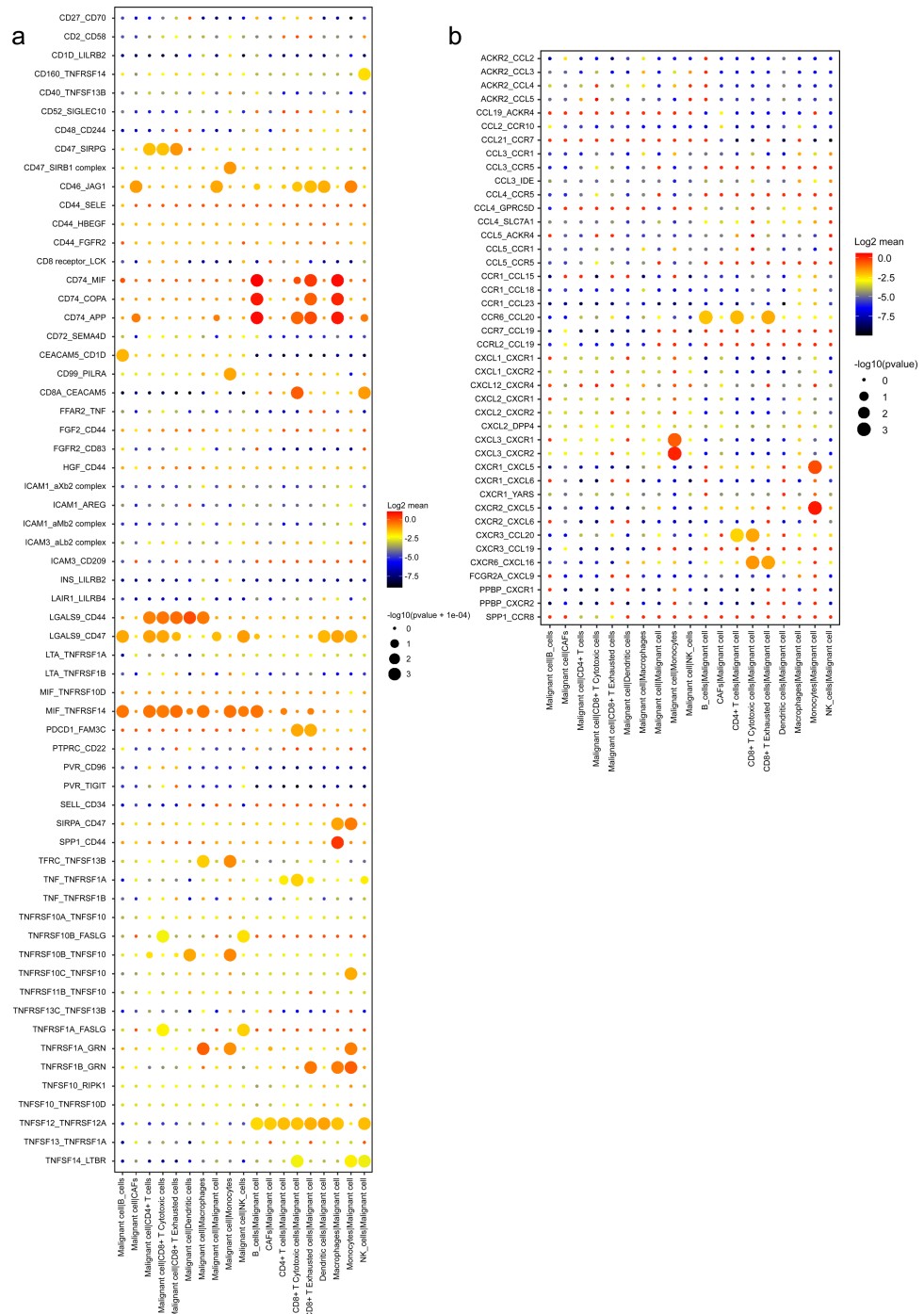

**Figure 7 Stimulatory and chemokine interations between malignant ductal cells and other clusters.** A summary of selected ligand–receptor interactions in malignant cells under (A) Stimulatory and (B) chemokine interactions.

CD274, CTLA4, PDCD1 and PDCD1LG2) (Fig. S10). In addition, it indicated that REG4 was almost an independent factor of immune infiltration. According to the Kaplan–Meier

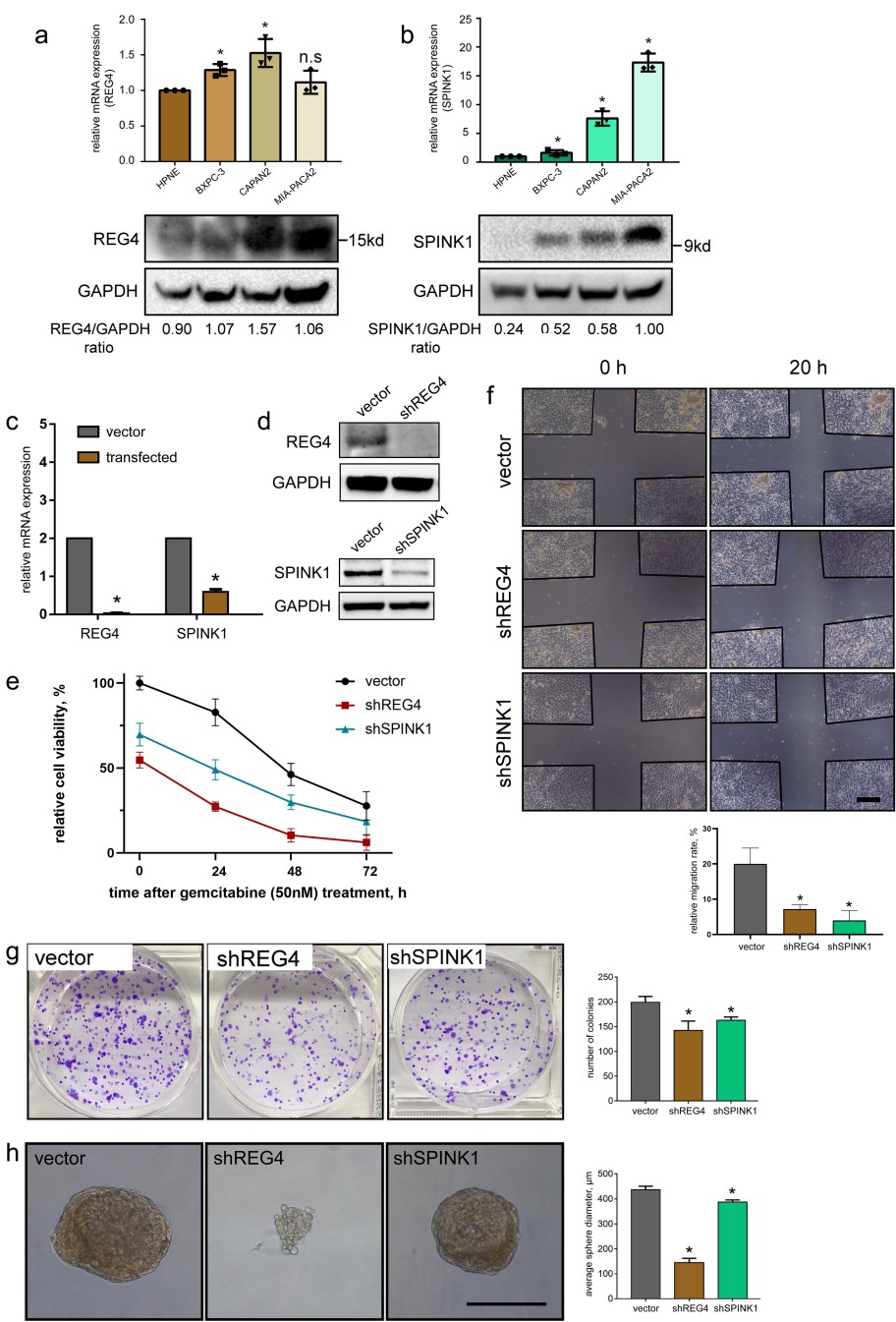

**Figure 8  Validation of gene expression and their potential effects on PDAC.** *In vitro* expression valida-
tion of potential prognostic genes. mRNA and protein expressions of (A) REG4 and (B) SPINK1 were ob-
tained with qRT-PCR and western blotting assays respectively, with HPNE as a control. Transfection ef-
fciency was validated by (C) mRNA and (D) protein levels. (E) Effects on MIA-PACA2 cell viability and
its sensitivity to gemcitabine (50 nM) after genetical manipulation of 2 genes. A series of functional ex-
periments including (F) wound healing, (G) colony formation and (H) sphere formation assays was con-
ducted to examine the effect of REG4 and SPINK1. Scale bar = 500μm. *$P < 0.05$ indicates statistical sig-
nificance. n.s indicates non-significance.

analysis, a negative but nonsignificant correlation between overall survival rate of PDAC patients and gene expression of REG4 and SPINK1 was detected respectively (Fig. S11).

Further, stable transfected MIA-PACA2 cells were established to confirm potential effects of these two genes on PDAC progression (Figs. 8C, 8D). We found knockdown of REG4 or SPINK1 do alter a series of cancer-related characteristics in PDAC. In specific, MIA-PACA2 cells transfected with shREG4 or shSPINK1 exhibited significant decrease in proliferation, drug resistance, migration, invasion, as well as CSC features (Figs. 8E–8H).

## DISCUSSION

PDAC is characterized by a high degree of intratumoral heterogeneity and aggressive features that constitutes the main obstacle to effective treatment (*Winter et al., 2012*; *Gobbi et al., 2013*; *Ryan, Hong & Bardeesy, 2014*; *Siegel et al., 2022*). Although profound achievements have been made for multiple solid tumors thanks to the advancement in single-cell technologies, it remains a challenging entity for studies in PDAC, due to the frequently low tumor cellularity and cellular heterogeneity in PDAC (*Kanda et al., 2012*; *Teng et al., 2015*; *Moffitt et al., 2015*; *Ho, Jaffee & Zheng, 2020*; *Steele et al., 2020*). There is no subtype that currently informs clinical decisions for PDAC yet (*Collisson et al., 2019*), so it is highly desirable to develop a *de novo* molecular taxonomy to guide treatment decisions.

Herein, we performed comprehensive analyses on a dataset of 16 primary PDAC lesions, accompanied with 3 adjacent tissues, at single-cell resolution, which revealed 6 major cell populations comprised of ductal cells and five stromal cellular types (myeloid cells, T cells, B cells, NK cells and CAFs). Malignant ductal cells were subclustered into five subpopulations with distinct transcriptomic and biological patterns, in line with previous results, highlighting substantial intratumoral heterogeneity in human primary PDACs (*Moffitt et al., 2015*; *Ho, Jaffee & Zheng, 2020*). Intensive crosstalk was found between malignant cell subtypes and stromal cells, which strongly overlaps with recent published PDAC scRNA-seq data (*Peng et al., 2019*; *Steele et al., 2020*; *Lee et al., 2021*). Meanwhile, we identified REG4 and SPINK1 as potential biomarkers for prognosis and therapeutic targets for PDAC.

Thanks to the development of novel transcriptomic techniques, especially single-cell sequencing, more different subtype classification schemes have been developed for PDAC, with majority agreement on a distinction between classical and basal-like phenotypes, which was originally proposed by Collisson and colleagues in 2011. The classical subtype usually presents well-differentiated subtypes of tumors characterized by epithelial gene expression, while mesenchymal genes are dominantly expressed in the basal-like tumor subtype (*Collisson et al., 2011*). The predominant discrepancy in metabolic features between classical and basal-like subtypes has been widely reported, with the classical subtype strongly associating with lipogenic phenotypes and the basal-like subtype preferentially wiring to glycolysis (*Daemen et al., 2015*; *Gutiérrez, Muñoz Bellvís & Orfao, 2021*). In addition, it has also been suggested that the basal-like subtype involves in a poorer prognosis and is less susceptible to chemotherapy compared with the classical subtype. As for our study, pathway analysis found known PDAC driver pathways, such as KRAS, MYC, MAPK

and WNT signaling pathways (*Australian Pancreatic Cancer Genome Initiative et al., 2012*; *Australian Pancreatic Cancer Genome Initiative et al., 2015*; *Witkiewicz et al., 2015*; *Raphael et al., 2017*), enriched in the five subtypes, which supported their tumor origin. Meanwhile, our results also correspond with the 'classical and basal-like' classification to a large extent. Specifically, we found representative pathways belonging to the classical subtype display specifically in subgroup 1 and 5, including steroid biosynthesis and cholesterol metabolism. While Subgroup 3 was predominantly enriched in hypoxia, KRAS signaling and EMT, sharing much more similarities with the basal-like features. Subgroup 2 presented a hybrid modality coexisting both classical (fatty acid metabolism and Notch pathway) and basal-like (glycolysis and KRAS signaling pathway) features. This result confirms recent notions that the molecular subtypes of PDAC cannot be directly recapitulated when it comes to individual cell populations analyzed at intratumoral levels (*Pompella et al., 2020*). In this regard, it is urgently necessary to identify PDAC exhibiting intermediate molecular features more precisely in order to reveal the great molecular heterogeneity of PDAC, as well as to exploit for therapies based on unique tumor subtypes. It also highlights the need to include an expanded cohort for further studies to make a firm conclusion.

It has been generally accepted that an elevated insulin level (hyperinsulinemia) indicates a common phenomenon in PDAC patients and a signal for poor clinical outcomes (*Zhang et al., 2019*; *Deng et al., 2022*). Insulin triggers signaling cascades implicated in tumorigenesis that promotes mitogenic activities (*Pollak, 2008*). Meanwhile, relevant researches indicated that insulin restored the capacity to instigate invasion and proliferation in pancreatic ductal cells with the introduction of KRAS mutation (*Boursi et al., 2017*; *Carreras-Torres et al., 2017*). While the failure in clinical trials aimed at insulin and relevant signaling pathways indicates that mechanisms by which insulin influences PDAC development are still unclear (*Zhang et al., 2019*). Notably, our study has strengthened the insulin-cancer notion by revealing functional enrichment in diabetes mellitus and insulin signaling pathways for subtype 1 and 2. Extensive explorations focused on insulin signaling pathways are needed to find novel druggable targets for PDAC therapy.

PDAC microenvironment is highly immunosuppressive, featured by a dense desmoplastic stroma that interferes with blood flow, inhibits drug delivery, and suppresses antitumor immunity (*Looi et al., 2019*; *Ho, Jaffee & Zheng, 2020*). Notably, unique immune-related signatures including TNF-$\beta$ signaling and EMT were detected in subgroup 3 that could be distinguished from other subtypes in our study. On the one side, TGF-$\beta$ blockage and EMT inhibition have been shown to reprogram the TME context and modify its immunologic conditions (*Gough, Xiang & Mishra, 2021*). On the other side, subgroup 2 and 3 also functioned extensively in immune regulation, involving various innate immune responses and immune cells including B cells, T cells and NK cells, which further contributed to immune escape of tumor cells.

Moreover, intimate cell–cell interactions among malignant cells and other stromal cells were suggested in our study as previously reported (*Wang et al., 2021*). It is possible that the resistance to tumor immunotherapy might be attributed to high levels of inhibitory receptor–ligand complexes (*Jiang et al., 2021*). Additionally, upregulated chemokine expressions, like CCL5 and CCL20 (*Romero et al., 2020*), might facilitate immune cells

recruited to tumors. Macrophages interacted most intensively with other cells (*Vitale et al., 2019*), suggesting their potential and promising abilities in anti-tumor treatments. Several studies have illustrated that CSCs are the primary source of tumorigenesis (*Peng et al., 2019*; *Nimmakayala et al., 2021*). As for our trajectory analysis, we found known CSC makers, such as ALDH1A1 (*Lee, Dosch & Simeone, 2008*; *Sergeant et al., 2009*), altered significantly during the transition to the malignant phenotype as well. Meanwhile, REG4 (*Hwang et al., 2020*) and SPINK1 (*Suzuki & Shimizu, 2019*), two of the potential differentially expressed genes validated to significantly upregulate during the trajectory, have been suggested to act as factors promoting CSC properties, which were also validated by sphere formation assay in our study (Fig. 8H). Moreover, we were intrigued to find impacts of both REG4 and SPINK1 on cell proliferation, drug resistance, migration and invasion abilities through functional validation assays. This finding suggests potential pivotal roles for REG4 and SPINK1 in PDAC progression and they may hold promise as potential therapeutic targets for PDAC in the future. We also found that multiple canonical oncogenic pathways, such as Notch and PTEN signaling pathways, were activated during trajectory progression. Based on our result, these pathways may be implicated in key molecular regulations in PDAC progression. However, these results in our analysis need more confirmation in future studies.

## CONCLUSIONS

Our understanding of the ductal cell subpopulations in PDAC offers new insights into evolving a novel framework for metabolic and immunologic therapies. REG4 and SPINK1 are expected to be promising prognostic markers for PDAC therapy. Further researches are needed to uncover the molecular mechanisms.

## ACKNOWLEDGEMENTS

We would like to give our sincere appreciation to the reviewers for their helpful comments on this article and the research group for the NCBI, which provided data for this collection.

### Funding
This work was supported by National Natural Science Foundation of China (No. 82072650) and the Natural Science Foundation of Zhejiang province, China (No. LQ21H160037). The funders had no role in study design, data collection and analysis, decision to publish, or preparation of the manuscript.

### Grant Disclosures
The following grant information was disclosed by the authors:
National Natural Science Foundation of China: 82072650.
Natural Science Foundation of Zhejiang province, China: LQ21H160037.

## Competing Interests

The authors declare there are no competing interests.

## Author Contributions

- Yutian Ji conceived and designed the experiments, performed the experiments, analyzed the data, prepared figures and/or tables, authored or reviewed drafts of the article, and approved the final draft.
- Qianhui Xu performed the experiments, analyzed the data, prepared figures and/or tables, and approved the final draft.
- Weilin Wang conceived and designed the experiments, authored or reviewed drafts of the article, and approved the final draft.

## Data Availability

The datasets analyzed for this study is available at GEO: GSE155698.

The original blots, codes for analysis and the qRT-PCR results are available in the Supplementary Files.

## Supplemental Information

Supplemental information for this article can be found online at http://dx.doi.org/10.7717/peerj.17350#supplemental-information.

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
