# Peer review of "Single-cell transcriptome reveals the heterogeneity of malignant ductal cells and the prognostic value of REG4 and SPINK1 in primary pancreatic ductal adenocarcinoma"

_PeerJ, doi:10.7717/peerj.17350_

## Round 0.1 · original submission · Major Revisions

Dear Dr. Ji and colleagues:

Thanks for submitting your manuscript to PeerJ. I have now received three independent reviews of your work, and as you will see, the reviewers raised some major concerns about the research (and the manuscript). Despite this, these reviewers are optimistic about your work and the potential impact it will have on research on cancer and cancer therapy. Thus, I encourage you to revise your manuscript, accordingly, considering all the concerns raised by the three reviewers.

The major criticism across reviews pertains to clarifying and justifying your methods, including ensuring that all information is available to effectively replicate your work. The reviewers would like to know why some datasets were excluded, while others might not be the best for comparative purposes. Your revision should strive to be more robust with these explanations and comparisons. Your manuscript also seems to fall short in effectively conveying the broader applicability to the community.

Please note that reviewer 3 kindly provided a marked-up version of your manuscript.

There are other minor problems pointed out by the reviewers, and you will need to address all of these and expect a thorough review of your revised manuscript by these same reviewers.

I agree with the concerns of the reviewers, and thus feel that their suggestions should be adequately addressed before moving forward. Therefore, I am recommending that you revise your manuscript, accordingly, considering all the issues raised by the reviewers.

I look forward to seeing your revision, and thanks again for submitting your work to PeerJ.

Good luck with your revision,

-joe

Reviewer 1 ·

Basic reporting

This study focused on pancreatic ductal adenocarcinoma (PDAC), a major contributor to cancer-related deaths with limited treatment options. The researchers used published scRNA-seq dataset to uncover heterogeneity and identify prognostic targets. They clustered 17,324 cells into five populations, defining a ductal cell cluster as malignant with high copy number variation. Malignant cells were further subtyped into five subgroups, each exhibiting specific immunologic and metabolic features. Trajectory analysis revealed differential expression of oncogenic pathways during tumor progression. Additionally, significant crosstalk between ductal cells and stromal cells was identified. Two genes, REG4 and SPINK1, were identified as promising therapeutic targets, as silencing them impaired various malignant properties of PDAC cells. Overall, the findings provide insights into PDAC heterogeneity and propose potential therapeutic targets.
The manuscript is well-written, but several questions need to be addressed:

Experimental design

The original dataset comprised 16 PDAC tissue samples and 3 adjacent normal pancreas samples; however, the authors opted to analyze only 10 PDAC samples. It would be beneficial for the authors to clarify the rationale behind this decision and consider utilizing the entire dataset for a more comprehensive comparison between tumor and normal tissues.

Validity of the findings

In figures 3e and S5c, the authors depict gene expression heatmaps on pseudotime trajectory; however, a significant portion of cells exhibit very low expression of the target genes. Bright cells seem to scatter somewhat randomly, making it challenging to validate the authors' conclusions. Further clarification or adjustments may be necessary to strengthen the reliability of their findings.
Figure 6g would benefit from the inclusion of quantifications. The authors should consider adding numerical data to provide a more precise representation of the results, contributing to the overall robustness and clarity of the findings.

Additional comments

NK cells are among the most abundant immune cell types identified in various tumors. The authors should provide an explanation for their absence in the dataset, addressing any technical or biological reasons that may account for this observation.
The authors should provide an explanation for the meaning of Branch Y-5 and Y-6 in figure 4d.

·

Basic reporting

no comment

Experimental design

The research question is well defined and experimental procedures well laid out.

Validity of the findings

This study on pancreatic ductal adenocarcinoma (PDAC) highlights its treatment challenges due to significant intratumoral heterogeneity. Using UMAP clustering on 10 primary PDAC lesions, Ji Y et al, identified five main cell types, including ductal and various stromal cells. It also revealed five distinct subpopulations of malignant ductal cells with unique transcriptomic patterns, supporting previous findings about PDAC's heterogeneity. The study noted important interactions between these cell types and identified REG4 and SPINK1 as potential prognostic biomarkers and therapeutic targets. The lack of a PDAC subtype for clinical decision-making underscores the need for a new molecular taxonomy to guide treatment.


1. Growth curve analysis over several days needs to be performed after gemcitabine treatment
2. Cell viability in stable cell lines needs to performed
3. What is the protein expression of the relative gene in the stable cell lines?
4. In Figure 6 panel h the different individual images are not labelled
5. Expression of REG4 and should be tested in primary and metastatic PDAC samples.
6. Typo in 469
7. In lines 469-473 the authors suggest using LGALS1 as a draggable target in PDAC. However apart from expression analysis, none of the data seems to suggest a link between LGALS1 and PDAC progression. Thus, this conclusion should be corrected and rewritten based on the data available.
Similarly, no data has been presented to speculate lines 487-491.
The discussion needs to be re-written with discussing the potential role of REG4 and SPINK1 in PDAC.

Additional comments

The discussion needs to be re-written. I have mentioned the specifics in part 3.

Reviewer 3 ·

Basic reporting

Their methodology and approach is valid. The English is clear but there are several grammatical errors through the manuscript that need to be corrected. The introduction and background show relevant context with references. Some of the figures are pixelated and need to be replaced.

Experimental design

Research question is valid, well stated and experimental approach is standard

Validity of the findings

The data is sound. Please see the PDF for more details.

Annotated reviews are not available for download in order to protect the identity of reviewers who chose to remain anonymous.

---

## Round 0.2 · Minor Revisions

Dear Dr. Ji and colleagues:

Thanks for revising your manuscript. The reviewer is very satisfied with your revision (as am I). Great! However, there are a few minor issues to address. Please address these ASAP so we may move towards acceptance of your work.

Best,

-joe

Reviewer 1 ·

Basic reporting

The authors have effectively tackled the majority of my prior concerns. However, I do have two additional inquiries regarding their latest analysis: the caption for Figure 2 implies that they conducted CNV analysis on 16 PDAC samples and 3 adjacent normal tissues. It would be beneficial for them to distinctly highlight or separate the ductal cells from tumor or normal tissues. We anticipate that ductal cells from normal tissue would exhibit minimal CNV events, which would also validate the accuracy of the CNV analysis.

The authors should also explain why the UMAP in Figure 2C look different from the UMAP in Figure 1A.

Experimental design

no comment

Validity of the findings

no comment

---

## Round 0.3 · accepted · Accept

Dear Dr. Ji and colleagues:

Thanks for again revising your manuscript. I now believe that your manuscript is suitable for publication. Congratulations! I look forward to seeing this work in print, and I anticipate it being an important resource for research on cancer and cancer therapy. Thanks again for choosing PeerJ to publish such important work.

Best,

-joe